# Spillover of a Tobamovirus from the Australian Indigenous Flora to Invasive Weeds

**DOI:** 10.3390/v14081676

**Published:** 2022-07-29

**Authors:** Weinan Xu, Hua Li, Krishnapillai Sivasithamparam, Dieu Thi Tran, Michael G. K. Jones, Xin Chen, Stephen J. Wylie

**Affiliations:** 1Plant Biotechnology Research Group (Virology), Western Australian State Agricultural Biotechnology Centre, Murdoch University, 90 South Street, Murdoch 6150, Australia; xuweinan0817@163.com (W.X.); perthmuzi@gmail.com (H.L.); krishnapillai.sivasithamparam@uwa.edu.au (K.S.); dieu2009@gmail.com (D.T.T.); m.jones@murdoch.edu.au (M.G.K.J.); 2Institute of Industrial Crops, Jiangsu Academy of Agricultural Sciences, Nanjing 210014, China

**Keywords:** virus spillover, emergence, wild-plant virus, tobamovirus, virus transmission

## Abstract

The tobamovirus yellow tailflower mild mottle virus (YTMMV) was previously reported in wild plants of *Anthocercis* species (family Solanaceae) and other solanaceous indigenous species growing in natural habitats in Western Australia. Here, we undertook a survey of two introduced solanaceous weeds, namely *Solanum nigrum* (black nightshade) and *Physalis peruviana* (cape gooseberry) in the Perth metropolitan area and surrounds to determine if YTMMV has spread naturally to these species. At a remnant natural bushland site where both solanaceous weeds and indigenous *Anthocercis* hosts grew adjacent to one another, a proportion of *S. nigrum* and *P. peruviana* plants were asymptomatically-infected with YTMMV, confirming spillover had occurred. Populations of *S. nigrum* also grow as weeds in parts of the city isolated from remnant bushland and indigenous sources of YTMMV, and some of these populations were also infected with YTMMV. Fruit was harvested from virus-infected wild *S. nigrum* plants and the seed germinated under controlled conditions. Up to 80% of resultant seedlings derived from infected parent plants were infected with YTMMV, confirming that the virus is vertically-transmitted in *S. nigrum*, and therefore infection appears to be self-sustaining in this species. This is the first report of spillover of YTMMV to exotic weeds, and of vertical transmission of this tobamovirus. We discuss the roles of vertical and horizontal transmission in this spillover event, and its implications for biosecurity.

## 1. Introduction

Tobamoviruses (family *Virgaviridae*, genus *Tobamovirus*) are amongst the most damaging viruses of horticultural crops, especially those seed-borne viruses of wide distribution [1]. As far as is known, the tobamovirus yellow tailflower mild mottle virus (YTMMV) is confined to wild plants in Western Australia and poses little risk to international horticulture. YTMMV was first identified in species of *Anthocercis* (family Solanaceae), including *A. littoria* (yellow tailflower) and *A. illicifolia* (holly-leaf tailflower), and in 14 other solanaceous plants indigenous to Western Australia [2,3].

Under experimental conditions, YTMMV was capable of systemically-infecting seedlings of 22 *Nicotiana* species, two cultivars of *Capsicum annuum* (bell pepper and jalapeno chili), one genotype each of *Solanum betaceum* (tamarillo), *S. lycopersicum* (tomato), *S. melongena* (aubergine), *S. nigrum* (black nightshade), *Physalis angulata* (balloon cherry), *P. peruviana* (cape gooseberry), and *P. philadelphica* (tomatillo) [4,5]. Symptoms ranged from asymptomatic to very mild mosaic in tomato to death in *C. annuum* and *N. benthamiana* accession RA-4 [5]. Neither the full natural host range nor the geographical range of YTMMV is known; most of the approximately 192 indigenous, cultivated, and naturalized exotic solanaceous species described from Western Australia have not been screened for YTMMV infection, nor have plants growing beyond the state of Western Australian. Its closest known relative, Scopolia mild mottle virus (SMMoV) was also identified from a wild plant, *Scopolia japonica* (Japanese belladonna, family Solanaceae), in Japan [6]. It is reasonable to expect the two viruses share a common ancestor in the solanaceous flora of the Austro–Asian region. Two cultivated *Capsicum* plants from a commercial market garden located near the northern town of Carnarvon were naturally infected with YTMMV [3], which indicated a potential for spillover into wild plants.

Here we screened wild populations of the widespread exotic solanaceous weed *Solanum nigrum* and one population of the less common weed *Physalis peruviana* from the Perth metropolitan region and surrounding areas for evidence YTMMV infection resulting from natural spillover from indigenous *Anthocercis* hosts, and investigated *S. nigrum* seedling plants for evidence of vertical transmission of the virus. 

## 2. Materials and Methods

### 2.1. Sample Collection

Leaf samples from 209 wild-growing plants of *Anthocercis illicifolia* (52), *A. littoria* (56) *Solanum nigrum* (92), *S. lycopersicum* (1), and *Physalis peruviana* (8) were collected from 11 sites (Appendix A) extending from the Perth City metropolitan area to the town of Cervantes located ~200 km to the north of Perth City (Figure 1) in Western Australia. Seven whole plants comprising five YTMMV-infected and two YTMMV-uninfected wild *S. nigrum* seedlings were removed from the field and transplanted in pots in a greenhouse where they were grown to maturity and allowed to self-pollinate to test for vertical (seed) transmission of YTMMV.

### 2.2. RNA Extraction, Amplification, and Sequencing

Total RNA was extracted from 20–100 mg leaf samples using TRIZOL solution (Invitrogen) according to the protocol of Chomczynski and Mackey [7]. The MyTaq™ One-Step RT-PCR system (Bioline) was used to synthesise cDNA and amplify fragments of the YTMMV coat protein (CP) gene in the presence of virus-specific forward and reverse primers (YTMMVCPF: 5′-GCTTAAAGAGCGAATTGATGAG; YTMMVCPR: 5′-CCATTGTAGTCTTGCACAGCAC). Cycling conditions were 25 cycles at 95 °C for 30 s, 55 °C for 40 s, 72 °C for 50 s in 1× GoTaq Green Mastermix (Promega). Of the 92 virus-positive plants identified by RT-PCR (Table 1 and Appendix A), amplicons of 48 were randomly selected for sequencing to confirm the presence of YTMMV and to identify sequence polymorphisms (Appendix A). Amplicons were sequenced directly using the same primers, and primer sequences were removed from the ends where present. To identify amplification-induced sequence misincorporations, independent cDNA-synthesis, amplification, and sequencing was done twice for each sample. Alignment of YTMMV partial CP gene (356 nt) sequences was done in ClustalW [8] and pairwise sequence analysis was done in Mega 11 [9] using the Neighbor-Joining method [10].

### 2.3. Mechanical Inoculation of YTMMV

Wild-collected leaf samples (1–2 g) were placed in a sterile mortar with 20 mL of 100 mM phosphate buffer (pH 7) and silicaceous powder and ground with a sterile pestle. The ground leaf solution was used to mechanically inoculate six-week-old seedlings of *N. benthamiana* accession KL-3 [4]. Mock-inoculated plants were treated as above but without the ground leaf tissue. Leaf tissue from a YTMMV-infected *S. betaceum* experimental host plant (YTMMV isolate “betaceum” originally derived in 2017 from an *A. littoria* plant growing near Lake Thetis) was the positive control used to inoculate KL-3 plants as above.

### 2.4. Vertical Transmission of YTMMV in Seed

Five wild YTMMV-infected and two YTMMV-uninfected *S. nigrum* plants were collected from the Leeming and Bertram sites and transplanted to a greenhouse and grown in isolation from one another and from other infected plants. Flowers were allowed to self-pollinate and fruits mature. Fruit was harvested and all the seeds collected from each plant were mixed, washed, and dried. Fifty-five seeds were randomly chosen from the seeds harvested from each plant and germinated together in a pot in the greenhouse. Twenty-one days after sowing, the percent seed germination, seedling height, and cotyledon length of each seedling that grew were recorded. Ten seedlings were harvested at random from each pot and RNA was extracted from each and tested individually for the presence of YTMMV by RT-PCR as above.

## 3. Results

### 3.1. YTMMV Spillover

In this study, 209 wild plant leaf samples of five solanaceous plant species from 11 sites around the Perth Metropolitan Region were collected to test for the presence of YTMMV. At only two sites consisting of eight plants, Canning river and Murdoch, was YTMMV not detected in any plant (Appendix A). At the other nine sites, YTMMV was detected in *A. illicifolia* (26 infected plants of 52 plants tested), *A. littoria* (15/56), *P. peruviana* (6/8), and *S. nigrum* (45/92) (Appendix A), suggesting that YTMMV is widespread in the Perth region of Western Australia. Indeed, 44% of plants randomly selected were infected with YTMMV. Symptoms were latent in *S. nigrum* and *P. peruviana* plants. Some plants of *A. illicifolia* at the Yanchep site exhibited apparent virus-like symptoms of leaf distortion, mottling, and chlorosis (Figure 2), but it is stressed that these symptoms may be unrelated to YTMMV infection as symptoms were not apparent in other infected plants of the same species or of *A. littoria*.

### 3.2. Sequences

RT-PCR amplicons were sequenced from 48 of the 92 YTMMV-positive plants detected. Sequencing confirmed the amplicons were of YTMMV and these were assigned accession codes at GenBank (Appendix A). When the sequences were trimmed of primer sequences, sequences trimmed to 356 nt and aligned, there was greater than 99% sequence identity. The sequences were divided into two groups by the presence of two polymorphic sites. A synonymous T to C substitution was present at nt 5876 (position of CP sequence after mapping to complete YTMMV sequence) in 19 virus isolates, while the remaining 29 isolates had a synonymous C to T substitution at nt 5939. Neighbor-Joining pairwise analysis confirmed the two groups with bootstrap support, which were labelled A and B (Appendix A). 

### 3.3. Vertical Transmission of YTMMV in S. nigrum

Seeds of YTMMV-infected *S. nigrum* plants were collected from self-pollinated plants collected from the wild at Leeming and Bertram sites. The seeds were germinated and a subset of seedlings was tested for the presence of YTMMV using the RT-PCR assay (as above). Approximately 68% of seedlings were infected with YTMMV (Table 2). Seed viability may be reduced by virus infection, but more work is required to confirm this. The growth of seedlings may not be significantly affected by vertically-acquired YTMMV infection, but, again, more work is required to confirm this. 

## 4. Discussion

We report for the first time that *S. nigrum* and *P. peruviana*, both invasive exotic species in Australia, are spillover hosts of YTMMV. *Solanum nigrum* originated in Eurasia, while *P. peruviana* originated in South America. Both are invasive weeds in many countries [11], and both can have annual or perennial lifecycles, depending on the climate. In *S. nigrum* at least, YTMMV is transmitted vertically. In suburban Perth, infected *S. nigrum* populations occurred at sites also occupied by natural YTMMV host (*Anthocercis* species) and at sites separated by several kilometres from populations of known indigenous host plants, indicating that infected seed has been transported from the original spillover site, possibly by fruit-eating birds. Fruits of both *S. nigrum* and *P. peruviana* are food sources for animals, including people. Birds especially are known to spread the seed of both species over long distances [12,13,14,15,16,17,18,19,20]. In Western Australia, a number of fruit-eating bird species could be responsible for spreading seeds of these weeds, including parrots, pigeons, wattlebirds, and several corvids. Subsequent maintenance of infection within *S. nigrum* populations is likely to be by seed transmission but possibly also by pollen-borne transmission.

Our observations suggest that vertical transmission by seed may be important in perpetuating infection between generations in indigenous hosts too, and that horizontal transmission by direct contact between plants (roots or leaves) and/or infected pollen could be less important. This is illustrated at the Ledge Point site, where a larger number of *A. littoria* plants co-occurred with a smaller number of *A. illicifolia* plants. All *A. illicifolia* plants were infected with YTMMV, while none of the *A. littoria* plants were infected, although both species are vulnerable to YTMMV and both species flower simultaneously. Many tobamoviruses are pollen and seed-borne [21]. It is probable that at Ledge Point, an infected ancestral *A. illicifolia* plant(s) has spread YTMMV vertically to its seedling progeny, which now exist at the site. Seed and pollen-borne transmission of YTMMV has not been shown in indigenous hosts, and this constitutes ongoing research. The YTMMV isolates at Ledge Point appear to be genetically homogeneous in *A. illicifolia*, as illustrated in the tree (Appendix A), which somewhat supports this hypothesis. On the other hand, the larger *A. littoria* population at Ledge Point was uninfected by YTMMV, and we propose that the ancestral founder plant(s) were uninfected, as its progeny remains today. Thus, horizontal transmission between the two *Anthocercis* species appears not to have occurred at Ledge Point. Both *Anthocercis* species flower in the June to November period, making pollen/virus transfer between the species theoretically possible. That horizontal transmission did not occur at this site between two potential host species provides an opportunity for further research into the relative importance of vertical versus horizontal transmission between indigenous hosts and how these factors influence the likelihood of spillover events.

In contrast to the Ledge Point *Anthocercis* sp. populations, horizontal transmission between species did occur at the Yanchep site. Infected *A. illicifolia* plants at Yanchep appears to be the source of the virus that spilled over to *S. nigrum* and *P. peruviana* plants growing there. *A. illicifolia* and *S. nigrum* plants growing adjacent to one another were simultaneously flowering (Figure 2), suggesting cross pollination as a potential spillover route. Pollinator-mediated transmission of other tobamoviruses is reported in cucumber green mottle mosaic virus (CGMMV) [22,23] and tomato brown rugose fruit virus (ToBRFV) [24]. At Yanchep, some adjacent plants were in physical contact, suggesting this as another potential spillover route. The Yanchep YTMMV isolates are mostly of the same genotype (Appendix A), with one exception, irrespective of whether the host was indigenous or exotic, again supporting a scenario of horizontal transmission between species.

Both species of weeds studied here are invasive in many agricultural regions of the world, coming into contact with economically-important crop species [25,26,27] such as potato, tomato, and capsicum. *S. nigrum* has been reported as a host for several viruses, including tomato yellow leaf curl geminivirus [28], pepino mosaic virus [29], potato virus Y, potato virus M [30], and nightshade curly top virus [31]. Less is known about *P. peruviana* as a reservoir of viruses. Further research is underway to determine if spillover of YTMMV from either *Anthocercis* plants or these weed species to solanaceous crops has occurred in the region. 

The means by which horizontal transmission of YTMMV can occur between *Anthocercis* species and *S. nigrum* and *P. peruviana* is unknown. A possible (untested) route is transmission by insects of YTMMV-infected *Anthocercis* pollen to the styles of *S. nigrum* and *P. peruviana* flowers where it germinates and grows down the stigma. Although genetic barriers to inter-genus fertilization exist, no such barrier exists for viruses, which can infect the recipient plant. Inter-species transmission of viruses was reported in the nepovirus cherry leafroll virus [32]. Other pollen-associated mechanisms of virus transmission reported are related to damage to the flower by the pollinator insect, thereby transmitting the virus through the wound [33,34,35]. 

## Figures and Tables

**Figure 1 viruses-14-01676-f001:**
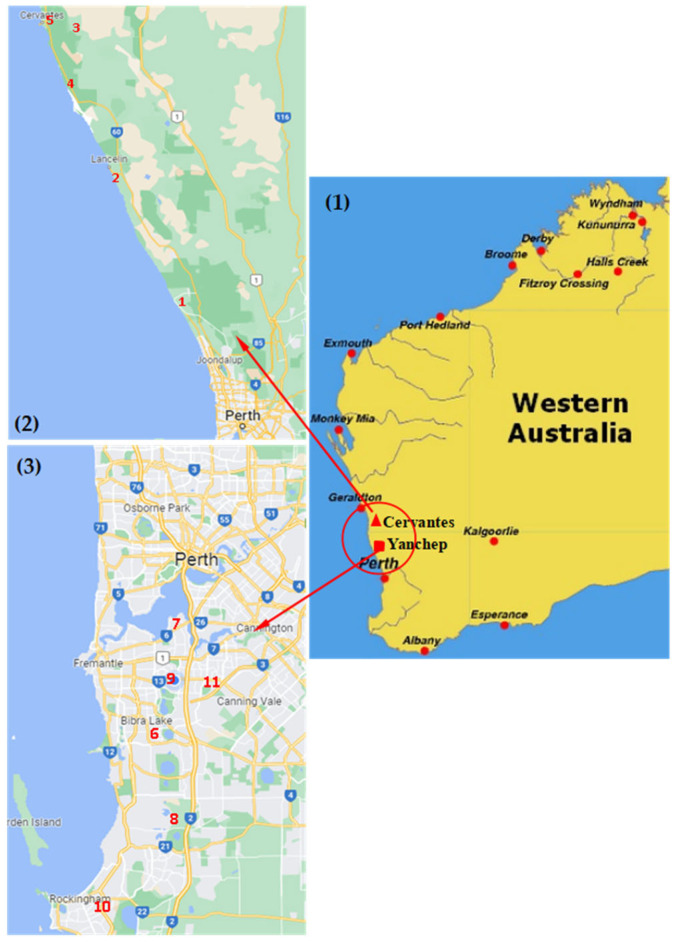
Locations of virus collection sites. (**1**) Western Australia showing the location of Perth, Yanchep, and Cervantes; (**2**) Sample collection sites to the north of the Perth metropolitan region: 1: Yanchep; 2, Ledge Point; 3, Lake Thetis; 4, dirt track in Cervantes region; 5, Cervantes sandhills; (**3**) Sample collection sites within the Perth metropolitan region: 6, Bibra Lake; 7, Canning River; 8, Bertram; 9, Murdoch; 10, Dixon Road; 11, Leeming.

**Figure 2 viruses-14-01676-f002:**
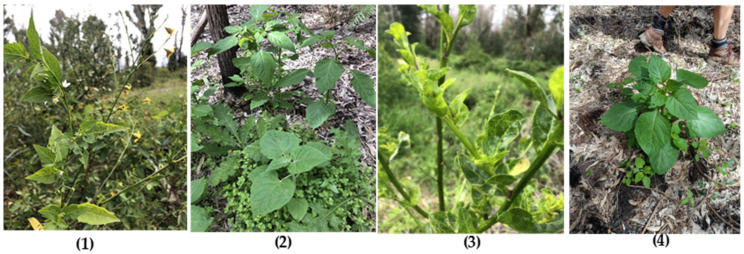
YTMMV-infected plants in the wild. (**1**) *Anthocercis illicifolia* (yellow flowers) and *Solanum nigrum* (white flowers) flowering adjacent to one another at Yanchep; (**2**) Infected *Physalis peruviana* (foreground) and *S. nigrum* (rear) plants growing adjacent to one another at Yanchep; (**3**) Leaf mosaic, distortion, and chlorosis on YTMMV-infected *A. illicifolia* plant at Yanchep. (**4**) Asymptomatic YTMMV-infected *S. nigrum* plants at the Bertram site.

**Table 1 viruses-14-01676-t001:** Natural infection with YTMMV at 11 sites. Numbers represent plants infected with YTMMV as a fraction of the total number of plants tested at each site.

Site/Species	*A. illicifolia*	*A. littoria*	*P. peruviana*	*S. nigrum*	*S. lycopersicum*
Yanchep	8/26	—	6/8	15/26	—
Ledge Point	5/5	0/28	—	—	—
Lake Thetis	13/21	—	—	—	—
Cervantes Dirt Track	—	3/5	—	—	—
Cervantes sandhills	—	12/23	—	—	—
Bibra Lake	—	—	—	2/11	0/1
Canning River	—	—	—	0/4	—
Bertram	—	—	—	11/17	—
Murdoch	—	—	—	0/4	—
Dixon Road	—	—	—	4/6	—
Leeming	—	—	—	13/24	—
Infected/tested	26/52	15/56	6/8	45/92	0/1

**Table 2 viruses-14-01676-t002:** Seed (vertical) transmission of YTMMV and germination and growth characteristics of *Solanum nigrum* seedlings.

Plant/Virus Code	Virus Presence ^a^	Seed Germination at 21 Days (%)	Seedling Stem Height at 21 Days (cm) ^b^	Seedling Cotyledon Length at 21 Days (cm) ^b^	Virus Transmission in 10 Seed
LM2	−	54/56 (96.4%)	1.77 ± 0.04 b	1.81 ± 0.04 b	0/10
LM8	−	50/55 (90.9%)	2.20 ± 0.04 a	1.98 ± 0.03 a	0/10
LM1	+	51/56 (91.1%)	1.33 ± 0.05 c	1.36 ± 0.05 d	7/10
LM3	+	50/55 (90.9%)	1.31 ± 0.03 c	1.16 ± 0.03 e	7/10
LM4	+	28/55 (50.1%)	1.41 ± 0.02 c	1.36 ± 0.05 d	8/10
LMB3	+	44/55 (80.0%)	1.69 ± 0.03 b	1.48 ± 0.02 c	7/10
BTA5	+	22/55 (40.0%)	1.18 ± 0.05 d	1.14 ± 0.02 e	5/10

^a^ − = virus not detected in parental plant, + = virus detected in parental plant by RT-PCR with species-specific primers for the coat protein gene. ^b^ Data are expressed as the mean ± standard error of three independent biological replicates. Different letters indicate significant differences at *p < 0.05*.

## Data Availability

Sequenced data in this study has been lodged at Genbank.

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
