# Peer review of "Spillover of a Tobamovirus from the Australian Indigenous Flora to Invasive Weeds"

_viruses, 2022, doi:10.3390/v14081676_

Round 1

Reviewer 1 Report

Recommended for publication 

Author Response

Thank you.

Reviewer 2 Report

This paper reports for the first time that Solanum nigrum and Physalis peruviana, both invasive exotic species in Australia, are hosts of YTMMV. Furthermore, the authors show that, in S. nigrum at least, YTMMV is transmitted vertically. The information although not bring sufficient novelty is interesting and thoroughly conducted and the information is valuable. I perceive no problems with the science in order to be published.

Specific comments:

- Please, clarify how horizontal transmission of the virus can occur between A. illicifolia and S. nigrum plants. Is cross-pollination possible between these species?

As far as I know, in the transmission of some viruses between species of different genera, the pollen from infected plants serve as a means for virus spreading through the direct intervention of insects, where would be necessary to cause lesions by feeding or otherwise so as to facilitate the exposure of cell protoplasm to infected pollen. This model has been suggested in:

Aramburu, J., et al (2010). Mode of transmission of Parietaria mottle virus. Journal of Plant Pathology, 92: 679-684.

Sdoodee R., Teakle D.S., 1993. Studies on the mechanism of transmission of pollen-associated Tobacco streak ilarvirus by Thrips tabaci. Plant Pathology 42: 88-92.

Greber R.S., Teakle D.S., Mink G.I., 1992. Thrips-facilitated transmission of Prune dwarf and Prunnus necrotic ringspot viruses from cherry pollen to cucumber. Plant Diseases 76: 1039-1041.

Greber R.S., Klose M.J., Teakle D.S., Milne J.R., 1991. High incidence of Tobacco streak virus in tobacco and its trans-mission by Microcephalothrips abdominatis and pollen from Ageratum houstonianum. Plant Disease 75: 450-452.

Hamilton et al., 1984. Survey of Prunus necrotic ringspot and other viruses contaminating the exine of pollen collected by bees. Canadian. Journal of Plant Pathology 6: 196-199.

Author Response

We added a paragraph to the discussion to address the reviewer's queries about inter-genus transmission. Four new references were also added to support the statements made in the new paragraph, as below:

The means by which horizontal transmission of YTMMV can occur between Anthocercis species and S. nigrum and P. peruviana is unknown. A possible (untested) route is transmission by insects of YTMMV-infected Anthocercis pollen to the styles of S. nigrum and P. peruviana flowers where it germinates and grows down the stigma. Although genetic barriers to inter-genus fertilization exist, no such barrier exists for viruses, which can infect the recipient plant. Inter-species transmission of viruses was reported in the nepovirus cherry leafroll virus [37]. Other pollen-associated mechanisms of virus transmission reported are related to damage to the flower by the pollinator, thereby transmitting the virus through the wound [38-40]. 

This manuscript is a resubmission of an earlier submission. The following is a list of the peer review reports and author responses from that submission.

Round 1

Reviewer 1 Report

This study reports the incidence of YTMMV in two indigenous wild species of Anthocercis, and in two exotic invasive weeds, in a restricted area of Western Australia around Perth. On the bases of incidence at ten different sites, on analyses of vertical transmission, and on nucleotide sequence comparisons, it is concluded that spill over occurred from the indigenous to invasive species. While these results could eventually produce a quite interesting manuscript, the present one does not allow to assess if  the results are sound to derive the conclussions.

Major comments:

1.    No information is given either in the Material and Methods section (2.3 and 2.5) or the Results (3.2) on the size of the CP amplicons sequenced. As only 6 polymorphic sites were detected (Table 3) it might be that the amplicon is too small to give information on YTMMV population diversity and structure, or else, that diversity is very small. Thus, this section of the results, including the phylogeny in Fig. 3 cannot be assessed.

2.    Fig. 3. Since the amplicon size is not known, and the NJ tress does not give any indication about the significance of nodes (bootstrap analysis), the occurrence of two clusters cannot be evaluated and, hence, neither if spill over occurred or not.

3.    Similarly, information on rates of seed transmission cannot be evaluated (section 3.4) as no information is given on the number of seeds/seedlings analysed per mother plant, nor on the number of mother plants.

4.    There is no indication of the potential interest of quantifying virus accumulation in different hosts, not why this analysis is done. In my opinion it adds nothing to this study. Having said that, the finding that accumulation was higher in N. benthamiana than in other hosts is to be expected, as N. benthamiana plants were kept in the green house, while the other hosts were in their natural setting. Most probably differences in virus accumulation can be due to environment rather than to host species.

Other comments.

5.    Scientific names of plant species should be written in italics (Legend to Fig. 2).

6.    Table 3 is not self-explanatory. It should be stated that isolates with a certain substitution are listed according to host.

Reviewer 2 Report

The manuscript by Xu et al. describes a survey of two solanaceous weeds, namely Solanum nigrum and Physalis peruviana in the Perth metropolitan area and surroundings. At one site, both weed species grew adjacent to YTMMV-infected Anthocercis plants, and both weed species were infected. Further, other S. nigrum populations located at distance from indigenous YTMMV hosts were infected, revealing long-distance spread, probably as infected seed by frugivorous birds, and maintenance within S. nigrum populations through vertical transmission of the virus. The work provides information on spillover of YTMMV hosts situation in Western Australia. The methodology is missing important aspect of plant disease confirmation i.e. Koch’s postulates. Furthermore, the presented data is insufficient for full-length research article as well as there is a need to write a comprehensive conclusion of this manuscript. The present manuscript is recommended for “Short Communication”